# Crystal Growth and Thermal Properties of Quasi-One-Dimensional van der Waals Material ZrSe_3_

**DOI:** 10.3390/mi13111994

**Published:** 2022-11-17

**Authors:** Youming Xu, Shucheng Guo, Xi Chen

**Affiliations:** Department of Electrical and Computer Engineering, University of California, Riverside, CA 92521, USA

**Keywords:** TMTCs, ZrSe_3_, quasi-1D materials, specific heat, thermal conductivity

## Abstract

ZrSe_3_ with a quasi-one-dimensional (quasi-1D) crystal structure belongs to the transition metal trichalcogenides (TMTCs) family. Owing to its unique optical, electrical, and optoelectrical properties, ZrSe_3_ is promising for applications in field effect transistors, photodetectors, and thermoelectrics. Compared with extensive studies of the above-mentioned physical properties, the thermal properties of ZrSe_3_ have not been experimentally investigated. Here, we report the crystal growth and thermal and optical properties of ZrSe_3_. Millimeter-sized single crystalline ZrSe_3_ flakes were prepared using a chemical vapor transport method. These flakes could be exfoliated into microribbons by liquid-phase exfoliation. The transmission electron microscope studies suggested that the obtained microribbons were single crystals along the chain axis. ZrSe_3_ exhibited a specific heat of 0.311 J g^−1^ K^−1^ at 300 K, close to the calculated value of the Dulong–Petit limit. The fitting of low-temperature specific heat led to a Debye temperature of 110 K and an average sound velocity of 2122 m s^−1^. The thermal conductivity of a polycrystalline ZrSe_3_ sample exhibited a maximum value of 10.4 ± 1.9 W m^−1^ K^−1^ at 40 K. The thermal conductivity decreased above 40 K and reached a room-temperature value of 5.4 ± 1.3 W m^−1^ K^−1^. The Debye model fitting of the solid thermal conductivity agreed well with the experimental data below 200 K but showed a deviation at high temperatures, indicating that optical phonons could substantially contribute to thermal transport at high temperatures. The calculated phonon mean free path decreased with temperatures between 2 and 21 K. The mean free path at 2 K approached 3 μm, which was similar to the grain size of the polycrystalline sample. This work provides useful insights into the preparation and thermal properties of quasi-1D ZrSe_3_.

## 1. Introduction

Transitional metal trichalcogenides (TMTCs) have attracted increasing research interest because of their unique quasi-one-dimensional (quasi-1D) crystal structures [1,2]. These materials contain two-dimensional (2D) layers, which are stacked in van der Waals (vdW) force between layers. In the 2D layers, they have stronger covalent bonding along one in-plane direction and weaker covalent bonding along the other. TMTCs have been extensively studied due to their interesting charge-density-wave (CDW) properties. The first discovery of CDW in TMTCs was reported in NbSe_3_ [3,4]. The phase transition occurred in NbSe_3_ was not accompanied by a structural transition, while a CDW formation was detected by electron diffraction. Afterward, CDW has been observed in TiS_3_ [5,6], HfTe_3_ [7], and ZrTe_3_ [8,9,10,11]. In addition to the CDW properties, TMTCs show other intriguing physical properties, such as superconductivity [12,13], optoelectronic behaviors [14,15,16,17,18,19], and thermoelectric properties [20,21,22]. 

Thermal transport properties are also important research topics for TMTCs. The single crystals of TMTCs exhibit anisotropic thermal transport properties due to their unique quasi-1D structures. Using a microthermal bridge method, Liu et al. discovered a high thermal conductivity (*κ*) along the chain axis in TiS_3_, twice the value along the other in-plane direction at room temperature, with 66% of thermal conductivity contributed by highly dispersive optical phonons [23]. Such dispersive optical phonons in TMTCs and corresponding anisotropic thermal conductivity were also observed in TaSe_3_ and ZrTe_3_ by theoretical calculations [24]. Recently, Yang et al. observed superdiffusive phonon transport in NbSe_3_ nanowires, revealing that the thermal conductivity followed a 1/3 power law dependence of the sample length [25]. This finding was attributed to drastic elastic stiffening along the 1D chain direction. As a result, phonons along the chain direction dominated thermal transport.

ZrSe_3_ is a semiconductor of the TMTC family, and it has a strong in-plane anisotropic structure. ZrSe_3_ crystallizes in the space group of P2_1_/m (No. 11) and can be synthesized by chemical vapor transport (CVT) [26]. Patel et al. studied the electrical and optical properties of single-crystal ZrSe_3_ and found the direct and indirect bandgap of ZrSe_3_ to be 1.1 and 1.47 eV, respectively [27]. Electrical resistivity data both parallel and perpendicular to *c*-axis decreased with increasing temperature, confirming its semiconducting nature. Osada et al. utilized Raman scattering to investigate the layer-dependent phonon properties of ZrSe_3_ [28]. When the number of layers decreased, the phonon vibration mode A_g_^3^, which reflected a quasi-1D structure, experienced a considerable blueshift. Wang et al. studied the anisotropic optical and optoelectronic properties of ZrSe_3_. The ZrSe_3_-based photodetector showed a wide photoresponse range with photoresponsivity of 11.9 mA W^−1^ at 532 nm [29]. Li et al. studied the effect of uniaxial strain along different crystal directions in ZrSe_3_ and discovered a strongly anisotropic exciton peak shift [30]. When the sample was strained along the *b*-axis, the exciton peak shift was much larger than along the *a*-axis. The first-principles studies suggested that the deformation along the *b*-axis modified the electronic bands of more orbitals compared with the deformation along the *a*-axis. Zhu et al. studied spin-orbit torques in ZrSe_3_/permalloy heterostructures [31]. When current was applied along the low-symmetry chain axis, an out-of-plane damping torque, corresponding to a large spin Hall conductivity, was detected in ZrSe_3_. In addition to these experimental studies, a recent theoretical study suggested that a large thermoelectric figure of merit *ZT* of 2.4 at 800 K can be achieved in monolayer ZrSe_3_ [32].

Compared with the active studies on electrical, optical and optoelectronic properties, the thermal properties of ZrSe_3_ have rarely been reported. In this work, we investigated the crystal growth and optical and thermal properties of ZrSe_3_. Millimeter-sized ZrSe_3_ single crystals were grown using the CVT method. These large crystals could be thinned down to microribbons by liquid-phase exfoliation. We further investigated the specific heat (*C_p_*) and thermal conductivity of polycrystalline ZrSe_3_ in the temperature range of 2–300 K. The analysis of specific heat data led to a Debye temperature of 110 K and an average sound velocity of 2122 m s^−1^. The thermal conductivity of ZrSe_3_ reached a peak value of 10.4 ± 1.9 W m^−1^ K^−1^ at 40 K and a room-temperature value of 5.4 ± 1.3 W m^−1^ K^−1^. The thermal conductivity was fitted via a Debye model, and the high-temperature deviation could be attributed to the optical phonon contribution to thermal conductivity. The phonon mean free path (MFP) calculated from the measured thermal conductivity increased with decreasing temperature, and approached a value of 3 μm at 2 K, which agreed well with the grain size of the polycrystalline sample.

## 2. Experimental Methods

### 2.1. Material Synthesis

The ZrSe_3_ crystals were synthesized via a CVT method [27]. The starting materials were zirconium powder (Zr, 100 mesh, purity >96%, Sigma Aldrich, Burlington, VT, USA), selenium powder (Se, 200 mesh, purity 99.999%, Alfa Aesar, Tewksbury, MA, USA), and iodine (I_2_, flakes, purity 99.8%, Sigma Aldrich, Burlington, VT, USA) as the transport agent. The Zr and Se powders with molar ratio of 1:3 were homogeneously mixed and sealed under vacuum in a closed quartz ampoule with an I_2_ concentration of 5 mg/mL. The ampoule was heated in a tube furnace at 1173 K for 120 h, followed by furnace cooling for 15 h. The as-synthesized product was shiny silver flakes. A dense ZrSe_3_ pellet for thermal property measurements was prepared by grinding the ZrSe_3_ crystals and cold-pressing the powder under 63 MPa at room temperature, followed by annealing at 1173 K for 24 h in a vacuum-sealed quartz tube. The liquid-phase exfoliation was performed by sonicating the ZrSe_3_ crystals in acetone for 4 h.

### 2.2. Material Characterization

The purity and crystal structure of the samples were characterized by a PANalytical Empyrean Series 2 powder X-ray diffraction (XRD) diffractometer (Malvern Panalytical, Malvern, UK) with a Cu Kα source (λ = 1.54 Å). The morphology of the samples was observed by a TESCAN Vega3 SBH scanning electron microscope (SEM) (TESCAN, Brno, Czech Republic) and a ThermoFisher Scientific Talos L120C transmission electron microscope (TEM) (ThermoFisher Scientific, Waltham, MA, USA). The pellet sample was cut into a typical dimension of 0.5 × 0.5 × 6 mm for the thermal conductivity measurement. The density (*ρ*) of the pellet sample was determined to be 4.27 g cm^−3^. A Quantum Design Physical Property Measurement System (PPMS) (Quantum Design, San Diego, CA, USA) was employed to measure the thermal conductivity along the direction perpendicular to the cold-pressing direction. The specific heat of the sample from 2 to 300 K was measured with the PPMS. The room-temperature Raman measurement was carried out with a HORIBA LabRam (HORIBA, Kyoto, Japan) using a 532 nm laser.

## 3. Results and Discussion

### 3.1. Phase and Microstructures

ZrSe_3_ exhibits a quasi-1D crystal structure with a monoclinic P2_1_/m space group (No. 11), as shown in Figure 1a. Each Zr atom is bonded to six Se atoms, forming an edge-sharing triangle prism along the *b*-axis (1D chains). The chains are stacked along the *a*-axis via a weaker covalent bond and form a 2D layer in the *ab* plane. The layers are further stacked by weak vdW forces along the *c*-axis. The room-temperature powder XRD pattern of ZrSe_3_ (Figure 1b) is consistent with the previously reported results [33], indicating that the pure ZrSe_3_ phase was formed by CVT. The corresponding lattice parameters are *a* = 5.415(7) Å, *b* = 3.753(4) Å, and *c* = 9.473(8) Å, with *α* = *γ* = 90° and *β* = 97.72°. In addition, a small amount of ZrO_2_ phase was observed, which could be attributed to the residual oxygen gas during the crystal growth.

Figure 2a,b show the SEM images of the as-synthesized ZrSe_3_ flakes. The lateral dimension of the flakes is about 1 mm. Microribbons with a width of about 1 μm could be observed at the edge of large flakes, as shown in Figure 2b. The quasi-1D microribbons are stacked in parallel, forming flat 2D layers through additional covalent Zr-Se bonding. The energy-dispersive spectroscopy (EDS) elemental mapping of constituent elements (Zr and Se) confirmed the chemical homogeneity of the flake, as displayed in Figure 2c. The quantitative EDS analysis was performed based on Zr Lα and Se Lα lines, and the stoichiometric ratio of Zr:Se was found to be approximately 1:2.9, indicating that a slight Se deficiency may have existed in the sample. The chalcogen element deficiency has also been reported in other quasi-1D transitional metal chalcogenides grown using the CVT method [34,35]. Further study is needed to quantify the Se vacancies in ZrSe_3_.

Due to the small exfoliation energy of ZrSe_3_ monolayers (0.37 J m^−2^) [36], ZrSe_3_ nanolayers were produced by mechanical exfoliation [30]. These findings motivated us to study the liquid-phase exfoliation of this compound. Figure 3a shows a typical microribbon of ZrSe_3_ with an in-plane dimension of 20 μm × 800 nm after liquid-phase exfoliation. The thickness of the microribbon was estimated to be less than 200 nm, verifying the effectiveness of liquid exfoliation to produce ZrSe_3_ nanolayers. The selected area electron diffraction (SAED) pattern (Figure 3c) could be indexed along the [001] zone axis, confirming the microribbon was along the *b*-axis, which was the chain direction.

For the thermal property measurements, we prepared a polycrystalline ZrSe_3_ pellet by cold-pressing the CVT single crystals followed by annealing in a vacuum. Figure 4a shows the SEM image of the fracture surface of the pellet sample after cold-pressing. Microribbons were randomly distributed within the bulk sample. The average grain size was found to be about 4 μm. As shown in Figure 4b, no compositional change could be observed in the XRD pattern of the sample after annealing in a vacuum, indicating its chemical stability.

### 3.2. Optical and Thermal Properties

The Raman spectrum of ZrSe_3_ at 300 K shows three characteristic peaks at 178, 235, and 302 cm^−1^ in Figure 5. According to a previous Raman study on bulk ZrSe_3_, three similar peaks at 178, 230, and 300 cm^−1^ were also observed and were assigned to A_g_^5^, A_g_^6^, and A_g_^8^ vibration modes, respectively [28]. Among them, A_g_^5^ and A_g_^6^ vibration modes correspond to the out-of-plane vibrations, and A_g_^8^ is due to the in-plane vibration mode. The A_g_^5^ mode consists of the movement of both Zr and Se atoms in the quasi-1D chains, while the A_g_^6^ and A_g_^8^ modes only consist of the movement of Se atoms.

The specific heat of ZrSe_3_ from 2 to 300 K is shown in Figure 6a. The specific heat monotonically increased with temperature up to 300 K, and slightly exceeded the Dulong–Petit limit of 0.304 J g^−1^ K^−1^ above 280 K [37]. According to the Debye model [38], specific heat can be fitted via the following equation at low temperatures:(1)$$\frac{C_{p}}{T}=\gamma +\frac{12\pi^{4}Nk_{B}}{5\theta_{D}{}^{3}}T^{2}$$
where *γ* is the electronic heat capacity coefficient, *k_B_* is the Boltzmann constant, *N* is the number of atoms per mole, *T* is the temperature, and *θ_D_* is the Debye temperature. C*_p_*/*T* versus *T*^2^ data below 9 K are shown in Figure 6b. The fitting led to a Debye temperature of 110 K with a sound velocity (*v_s_*) of 2122 m s^−1^, as listed in Table 1 together with other measured physical properties of ZrSe_3_. The obtained data are in good agreement with those from a previous study on ZrSe_3_ [39], where the reported Debye temperature and sound velocity were found to be 110 K and 2140 m s^−1^, respectively. The electronic heat capacity coefficient of ZrSe_3_ is negligible, in agreement with its semiconductor nature.

Figure 6c shows the measured thermal conductivity of the cold-pressed ZrSe_3_ sample in comparison with the data for a ZrTe_3_ polycrystal [40]. ZrTe_3_ has the same crystal structure as ZrSe_3_ but with larger lattice parameters. The thermal conductivity of ZrSe_3_ shows a clear peak at 40 K, while the thermal conductivity of ZrTe_3_ exhibits a broad plateau in the temperature range of 120−300 K. ZrSe_3_ shows a stronger temperature dependence of thermal conductivity at low temperatures compared with ZrTe_3_, indicating that the thermal conductivity of ZrSe_3_ is less affected by boundary scattering and defect scattering than ZrTe_3_. The resistivity of the ZrSe_3_ polycrystalline sample exceeds the measurement limit of the PPMS. From a previous study, the resistivity of ZrSe_3_ single crystal was reported to be 143.9 Ω cm at room temperature [27]. According to the Wiedemann–Franz law, with a Lorentz number of 2.44 × 10^−8^ V^2^ K^−2^, the calculated electronic thermal conductivity (*κ*_E_) at 300 K is 5.1 × 10^−6^ W m^−1^ K^−1^, which is negligible compared with lattice thermal conductiity (*κ*_L_).

At higher temperatures where phonon–phonon scattering dominates, the thermal conductivity of ZrSe_3_ is lower than that of ZrTe_3_, with values of 5.4 W m^−1^ K^−1^ and 7 W m^−1^ K^−1^ at 300 K, respectively. However, because the Te atom is heavier than the Se atom, the phonon spectrum of ZrTe_3_ should be narrower than that of ZrSe_3_, leading to smaller phonon velocities and stronger phonon–phonon scattering in ZrTe_3_. Thus, the thermal conductivity of ZrTe_3_ should be lower than that of ZrSe_3_, which seems to contradict our results. This discrepancy may have been caused by different synthesis parameters for the two samples, resulting in different porosities and texture effects. The calculated thermal conductivity of ZrTe_3_ along chain, cross-chain, and cross-plane directions are 9.6, 3.9, and 2.3 W m^−1^ K^−1^ at 300 K, respectively [24]. The polycrystalline ZrTe_3_ was reported to have texture effects with preferred orientation along the chain axis [40], showing an experimental thermal conductivity close to the calculated value along the chain direction. In addition, a small amount of ZrO_2_, observed by the XRD study, could enhance phonon scattering, and thus possibly decrease the thermal conductivity of ZrSe_3_.

### 3.3. Thermal Transport Analysis

The Debye model was used to analyze the thermal transport in ZrSe_3_. Before the Debye model fitting, the measured thermal conductivity needed to be corrected for porosity (*f*) because the cold-pressed sample was not dense enough and contained voids, as can be seen from Figure 4a. The porosity of the sample can be calculated from the following equation
(2)$$f=\left(1-\frac{\rho }{\rho_{theor}}\right)\times 100\%$$
where *ρ_theor_* is the theoretical density of ZrSe_3_. The porosity of ZrSe_3_ was calculated to be 18%.

According to the effective medium theory [42], the solid thermal conductivity ($\kappa_{s}$) is related to the measured thermal conductivity as
(3)$$\kappa_{s}=\kappa {\left(\frac{2-2f}{2+2f}\right)}^{-1}$$

The obtained $\kappa_{s}$ of ZrSe_3_ is presented in Figure 6c. We fit the solid thermal conductivity of ZrSe_3_ from 2 to 200 K using the following Debye model [43],
(4)$$\kappa_{s}=\frac{k_{B}}{2\pi^{2}v_{s}}{\left(\frac{k_{B}T}{\hbar }\right)}^{3}\int_{0}^{\theta_{D}/T}\frac{x^{4}e^{x}}{\tau_{c}^{-1}{\left(e^{x}-1\right)}^{2}}dx,$$
where ℏ is the reduced Planck constant, *x* = ℏ*ω*/*k_B_T*, *ω* is phonon frequency, and *τ_c_* is the lattice relaxation rate. The lattice relaxation rate consists of boundary scattering (*τ_B_*), point defect scattering (*τ_D_*) and Umklapp scattering (*τ_U_*) contributions and can be expressed as
(5)$$\tau_{c}^{-1}=\tau_{B}^{-1}+\tau_{D}^{-1}+\tau_{U}^{-1}$$

The relaxation rates for boundary scattering, point defect scattering, and Umklapp scattering, respectively, are given by
(6)$$\tau_{B}^{-1}=\frac{v_{s}}{L},\;\tau_{D}^{-1}=A\omega^{4},\;\tau_{U}^{-1}=B\omega^{2}Te^{-\frac{\theta_{D}}{3T}}$$
where *L* is the average grain size; and *A* and *B* are prefactors for point defect scattering and Umklapp scattering, respectively. Figure 6c shows the results of fitting the thermal conductivity to the Debye model. The obtained fitting parameters were *L* = 5.0 μm, *A* = 4.8 × 10^−42^ s^3^, and *B* = 2.1 × 10^−18^ s K^−1^. The obtained *L* value was consistent with the average grain size of about 4 μm from the SEM study (Figure 4a).

Extrapolating the fitting toward *T* > 200 K led to a deviation between the calculation and experimental data. Such a deviation could be attributed to the contribution of optical phonons to κ, which is not considered in the Debye model [44,45]. Debnath et al. calculated the phonon dispersion and thermal conductivity of ZrTe_3_ [24]. Several optical phonon modes were highly dispersive with large velocities. As a result, 33% of the lattice thermal conductivity was carried by optical phonons at room temperature. Similarly, Mortazavi et al. calculated the phonon dispersion of monolayer ZrSe_3_, and dispersive optical modes were also shown in the phonon dispersion [36]. These theoretical results are consistent with our findings, suggesting that the optical phonons in ZrSe_3_ can also contribute to thermal transport substantially.

In order to better understand the phonon transport in polycrystalline ZrSe_3_, acoustic phonon MFP can be calculated using the solid thermal conductivity as [46,47]
(7)$$l=\kappa_{s}/\left[\frac{k_{B}}{2\pi^{2}v_{s}^{2}}{\left(\frac{k_{B}T}{\hbar }\right)}^{3}\int_{0}^{\theta_{D}/T}\frac{x^{4}e^{x}}{{\left(e^{x}-1\right)}^{2}}dx\right]$$

The calculated MFP of ZrSe_3_ up to 21 K is shown in Figure 6d. As temperature decreases, the calculated acoustic phonon MFP increases and reaches 3 μm at 2 K, consistent with the findings of the SEM study.

Furthermore, the minimum thermal conductivity (*κ_min_*) of ZrSe_3_ can be calculated according to the model developed by Cahill et al. [41] with the following equation:(8)$$\kappa_{min}={\left(\frac{\pi }{6}\right)}^{1/3}k_{B}n_{A}^{2/3}v_{s}{\left(\frac{T}{\theta_{D}}\right)}^{2}\int_{0}^{\theta_{D}/T}\frac{x^{3}e^{x}}{{\left(e^{x}-1\right)}^{2}}dx$$
where *n_A_* is the density of atoms. The *κ_min_* of ZrSe_3_ was found to be 0.43 W m^−1^ K^−1^ at 300 K, which is less than one-tenth of the measured value. As such, it is expected that the thermal conductivity of ZrSe_3_ can be further suppressed by nanostructuring [48] or defect engineering [49] for thermoelectric applications.

## 4. Conclusions

We report the crystal growth and thermal properties of quasi-1D vdW material ZrSe_3_. Millimeter-sized ZrSe_3_ flakes were grown by the CVT method. Due to the weak vdW bond along the *c*-axis and relative weak covalent bond along the *a*-axis, the flakes could be exfoliated into microribbons using the liquid-phase exfoliation method. The cold-pressed ZrSe_3_ sample exhibited a maximum thermal conductivity of 10.4 ± 1.9 W m^−1^ K^−1^ at 40 K and a room-temperature value of 5.4 ± 1.3 W m^−1^ K^−1^. The thermal transport analysis showed good agreement between the experimental data and Debye model fitting below 200 K, suggesting that the phonon transport in the polycrystalline sample was dominated by grain boundary, point defect, and Umklapp scattering. The high-temperature deviation between the fitting and experimental data could be attributed to the contribution of optical phonons. Combining the effective medium theory and Debye model, the acoustic phonon mean free path was calculated to be 3 μm at 2 K, consistent with the SEM observation. In addition, the analysis of low-temperature specific heat led to a Debye temperature of 110 K and an average sound velocity of 2122 m s^−1^. This study provides the first experimental investigation of thermal transport in ZrSe_3_ as well as preparation of ZrSe_3_ nanostructures using liquid-phase exfoliation, which can enable novel applications based on quasi-1D ZrSe_3_.

## Figures and Tables

**Figure 1 micromachines-13-01994-f001:**
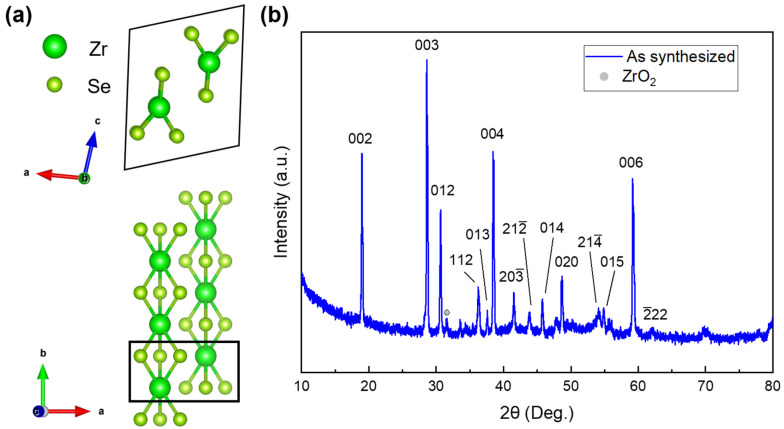
(**a**) Crystal structure of ZrSe_3_. Each Zr atom bonds to six neighbor Se atoms, forming a triagonal prism. (**b**) Indexed powder XRD pattern of the ZrSe_3_ sample.

**Figure 2 micromachines-13-01994-f002:**
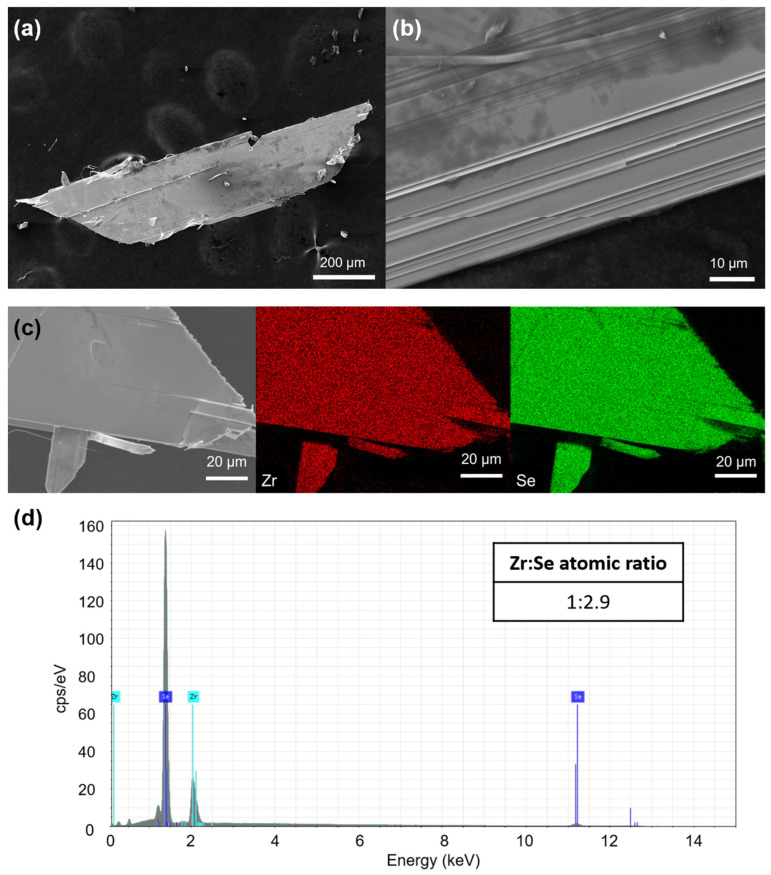
(**a**,**b**) SEM images of ZrSe_3_ after CVT. (**c**) EDS elemental mapping of Zr and Se in a ZrSe_3_ flake. (**d**) EDS spectrum of ZrSe_3_.

**Figure 3 micromachines-13-01994-f003:**
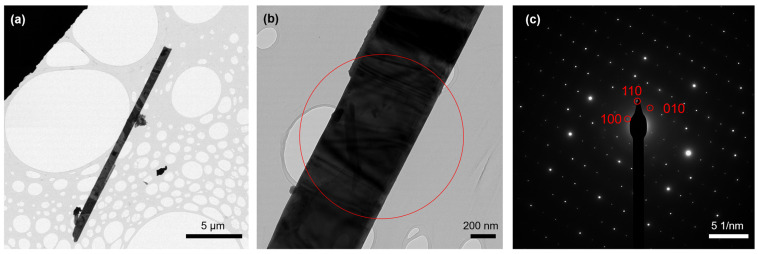
(**a**,**b**) TEM images of the ZrSe_3_ sample after liquid-phase exfoliation. The circle in (**b**) indicates the region for electron diffraction. (**c**) The corresponding SAED pattern obtained along [001] zone axis.

**Figure 4 micromachines-13-01994-f004:**
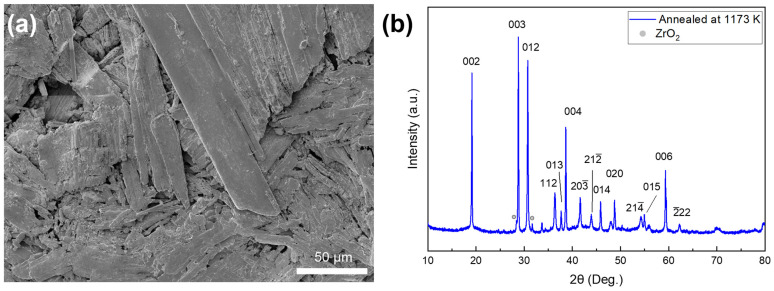
(**a**) SEM image of the ZrSe_3_ sample after cold-pressing showing the fracture surface perpendicular to the press direction. (**b**) Powder XRD pattern of ZrSe_3_ after annealing at 1173 K in vacuum.

**Figure 5 micromachines-13-01994-f005:**
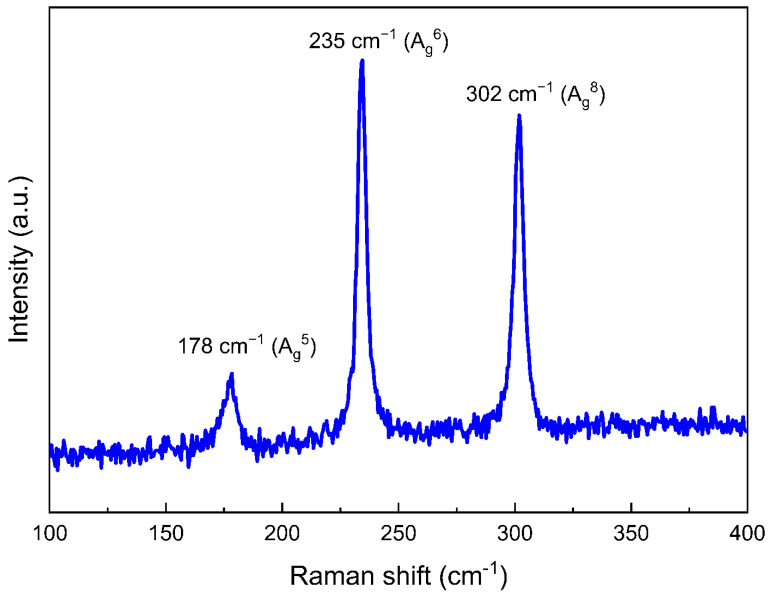
Raman spectrum of ZrSe_3_ at 300 K.

**Figure 6 micromachines-13-01994-f006:**
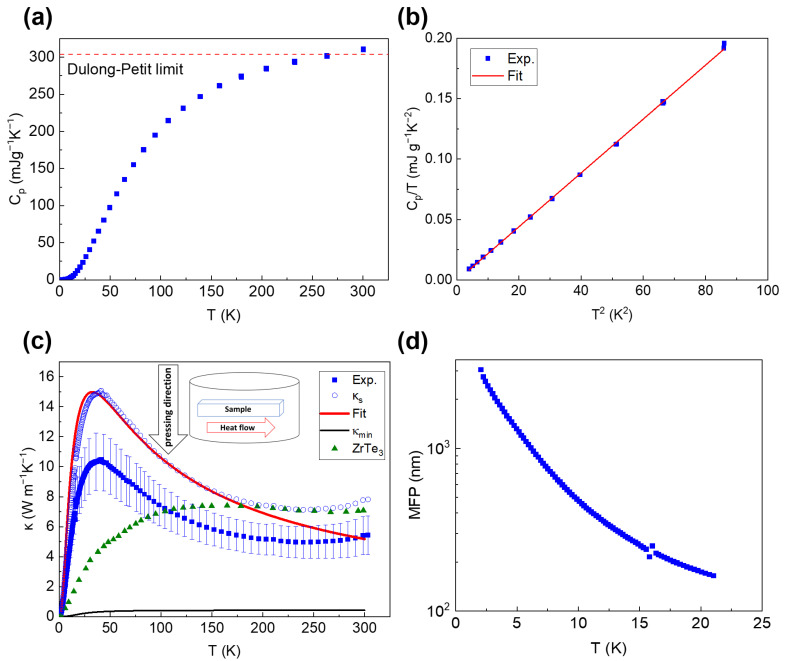
(**a**) Specific heat as a function of temperature for ZrSe_3_. The high-temperature limit was calculated using the Dulong–Petit law. (**b**) C*_p_*/*T* versus *T*^2^ at low temperatures. (**c**) Thermal conductivity of ZrSe_3_ as a function of temperature. The solid thermal conductivity was calculated by correcting the porosity effect. The experimental data for ZrTe_3_ are included for comparison [40]. The solid thermal conductivity was fitted using the Debye model. The minimum thermal conductivity was calculated via the Cahill model [41]. The inset of (**c**) is a schematic illustration of the thermal conductivity measurement direction. Reprinted/adapted with permission from Ref. [40]. Copyright 2019, Elsevier. (**d**) Phonon MFP of the cold-pressed sample as a function of temperature below 21 K.

**Table 1 micromachines-13-01994-t001:** Experimentally measured physical properties of ZrSe_3_.

Sample	*ρ* (g cm^−3^)	*C_p_* (J g^−1^ K^−1^)	*θ_D_* (K)	*v*_s_ (m s^−1^)	*κ* (W m^−1^ K^−1^)	*κ*_max_ (W m^−1^ K^−1^)
ZrSe_3_	4.27	0.311 (300 K)	110	2122	5.4 ± 1.3 (300 K)	10.4 ± 1.9 (40 K)

## Data Availability

The data presented in this study are available on request from the first author.

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
