# Peer review of "Crystal Growth and Thermal Properties of Quasi-One-Dimensional van der Waals Material ZrSe3"

_micromachines, 2022, doi:10.3390/mi13111994_

Round 1
Reviewer 1 Report
The manuscript, titled “Crystal Growth and Thermal Properties of Quasi-one-dimesional van der Waals Material ZrSe3” by Y. Xu, study the growth by CVD method and basic low-temperature thermal properties of ZrSe3, one promising material of TMTCs family. Authors report an insightful investigation on its crystal morphology and thermal transport below 300K. The fitting results report the Debye temperature of 100K, sound velocity of 2122 m/s, and thermal conductivity of 5.4W/mK at room temperature, which could provide the basic parameters of thermal properties of ZrSe3 for readers. I believe that the manuscript satisfies the criteria of the journal. However, some issues should be addressed before considering to be published in Micromachines.
1) Figure 1b shows that there is a small amount of ZrO2 phase in the sample, does the second phase affect the measurement results for thermal properties?
2) Authors report that the porosity of cold pressed ZrSe3 sample is around 18%, readers might be doubt about the fitting results of heat capacity and thermal conductivity, since the Debye model theoretically be applied in perfect solid materials.
3) The measurement of thermal conductivity is perpendicular to the cold-pressing direction, however, the SEM image in figure 4 shows the anisotropy of texture structure for polycrystalline ZrTe3 sample. Readers might be confused on the analysis of thermal conductivity data by using the average sound velocity instead of anisotropic velocity.
Author Response
Point 1: Figure 1b shows that there is a small amount of ZrO2 phase in the sample, does the second phase affect the measurement results for thermal properties?
Response 1: Yes, the second phase could enhance the phonon scattering, thus possibly decrease the thermal conductivity. We have expanded the discussion on page 8 as follows:
“A small amount of ZrO2, observed by the XRD study, could enhance phonon scattering, and thus possibly decrease the thermal conductivity of ZrSe3”
Point 2: Authors report that the porosity of cold pressed ZrSe3 sample is around 18%, readers might be doubt about the fitting results of heat capacity and thermal conductivity, since the Debye model theoretically be applied in perfect solid materials.
Response 2: Thank you for pointing this out. We have determined the solid thermal conductivity of the sample based on the porosity. We have then performed a new Debye model fitting to the solid thermal conductivity. The obtained grain size is 5.0 μm and the prefactors for point defect scattering and Umklapp scattering are A = 4.8Í10-42 s3, and B = 2.1Í10-18 s K-1, respectively. The modified fitting results are shown in Fig. 6c.
Regarding the specific heat, the porosity doesn’t affect the analysis since the specific heat was measured in the unit of J g-1K-1.
Point 3: The measurement of thermal conductivity is perpendicular to the cold-pressing direction, however, the SEM image in figure 4 shows the anisotropy of texture structure for polycrystalline ZrTe3 sample. Readers might be confused on the analysis of thermal conductivity data by using the average sound velocity instead of anisotropic velocity.
Response 3: As can be seen from the SEM image in Fig. 4, the flakes are aligned in different directions, indicating that the cold-pressing process doesn’t result in obvious texture for ZrSe3, possibly due to the low pressure applied in the cold-pressing (63 MPa). As such, it is expected that the cold-pressed pellet doesn’t show a strong anisotropy in thermal conductivity. Also, the obtained pellet sample is not thick enough to perform the thermal conductivity measurement along the pressing direction.
A similar discussion has been added on page 5 of the revised manuscript.

Reviewer 2 Report
In this study, the authors synthesized ZrSe3 with a quasi-one-dimensional crystalline structure by chemical vapor transport method and characterized its thermophysical properties from thermal conductivity measurements and Raman measurements. From the temperature dependence of the thermal conductivity of bulk ZrSe3, the mean free path of phonons was estimated to be 3 μm, which is almost consistent with the grain size of the polycrystalline sample. These research results are highly significant in terms of novelty and scientific significance, and are judged to be suitable for publication in Micromachines with some modifications and additions.
However, some modifications are necessary because of the lack of information on the synthesis conditions and the somewhat insufficient explanation of the orientation and thermal conductivity anisotropy of the measured samples.
1. The purity of the elements used in the synthesis should be stated.
2. Have the composition and crystal structure of the ZrSe3 pellets annealed in vacuum been confirmed by SEM-EDS and XRD? Have you confirmed that there is no compositional change or Se defects?
3. Is the electrical resistivity of the sample really high after annealing? Is the electronic thermal conductivity estimated from the Wiedemann-Franz law sufficiently small compared to the lattice thermal conductivity?
4. ZrSe3 is a highly anisotropic material, but is there any anisotropy in the thermal conductivity? Is it possible that the thermal conductivity of ZrSe3 is apparently lower than that of ZrTe3 due to the difference in anisotropy? The author needs to explain in which direction the heat flow was directed on the pellet in the thermal conductivity measurement.
Reviewer 3 Report
This study is largely well carried out, and it is suitable to be published in Micromachines. The manuscript can be accepted after a minor revision. Please see below the necessary revisions.
1. Subscript and superscript errors in the title and abstract (example: ZrSe3, 0.311 J g-1 K-1).
2. Please insert the atomic weight percentages of Zr and Se (quantitative EDS analysis) in Fig. 2d.
3. Unit of density is wrong, g cm-1? (line 108)
4. Any refinement (e.g. Rietveld) done on the XRD (Fig.1b) to obtain the lattice parameters? Please provide the refinement parameters and related information.
